# Yogurt as an Alternative Ingredient to Improve the Functional and Nutritional Properties of Gluten-Free Breads

**DOI:** 10.3390/foods9020111

**Published:** 2020-01-21

**Authors:** Carla Graça, Anabela Raymundo, Isabel Sousa

**Affiliations:** LEAF—Linking Landscape, Environment, Agriculture and Food, Research Center of Instituto Superior de Agronomia, Universidade de Lisboa, Tapada da Ajuda, 1349-017 Lisboa, Portugal; carlalopesgraca@isa.ulisboa.pt (C.G.); anabraymundo@isa.ulisboa.pt (A.R.)

**Keywords:** gluten-free bread, yogurt, rheology

## Abstract

Absence of gluten in bakery goods is a technological challenge, generating gluten-free breads with low functional and nutritional properties. However, these issues can be minimized using new protein sources, by the addition of nutritional added-value products. Fresh yogurt represents an interesting approach since it is a source of protein, polysaccharides, and minerals, with potential to mimic the gluten network, while improving the nutritional value of gluten-free products. In the present work, different levels of yogurt addition (5% up to 20% *weight*/*weight*) were incorporated into gluten-free bread formulations, and the impact on dough rheology properties and bread quality parameters were assessed. Linear correlations (R^2^ > 0.9041) between steady shear (viscosity) and oscillatory (elastic modulus, at 1 Hz) values of the dough rheology with bread quality parameters (volume and firmness) were obtained. Results confirmed that the yogurt addition led to a significant improvement on bread quality properties, increasing the volume and crumb softness and lowering the staling rate, with a good nutritional contribution in terms of proteins and minerals, to improve the daily diet of celiac people.

## 1. Introduction

Celiac disease is an immune enteropathy caused by the ingestion of gluten in genetically susceptible individuals, and it is estimated to affect about 1–3% of the population worldwide [1].

Currently, the gluten-free (GF) diet is the only option for people suffering from gluten-related disorders. For other reasons, a high number of nonceliac individuals are also adopting this diet. Subsequently, there has been a significant increase of research work to improve the functional and nutritional properties of GF products [2,3]. However, several studies stated that the consumers remain unsatisfied with the quality of the GF products in the market [4], highlighted their insufficiency on overall appearance and nutritional values, compared to the gluten-containing counterparts [5,6].

Gluten-free products, especially breads being mainly based on refined flours and starches, are generally characterized by poor technological quality attributes, including dry, crumbling texture, color, and mouth feel [7], and undergo fast staling [8].

In terms of nutritional profile, normally they present deficient values of protein and minerals and higher carbohydrates and fat content than recommended [9].

Rice flour has been widely proposed as an alternative for making gluten-free breads (GFB) due to its hypoallergenic protein, soft taste, and white color [10]. However, rice-based bread presented low quality attributes, in terms of volume and hard crumb [11]. Alternative flours from pseudocereal sources, such as amaranth, buckwheat, quinoa, sorghum, and teff, have been applied to improve the nutritional profile of the GFB [12].

In fact, gluten possesses unique viscoelastic properties which are crucial for the water-holding capacity of the dough and gas retention during fermentation [13]. To mimic this structure-building potential, hydrocolloids such as hydroxy-propril-methyl-cellulose, carboxy- or methyl-cellulose, locust bean, guar gum, and xanthan gum, as well as enzymes [8,14], are currently used to improve the viscoelastic properties and technological quality of the end-products [15]. Although these functional additives contribute to the gas retention during the fermentation process, improving the bread volume, they can be insufficient in terms of desirable bread texture and nutritional value.

Addition of new protein sources, e.g., dairy dry powders incorporation (skim milk powder, dry milk, whey protein concentrate) was shown to improve the volume, appearance, and sensory aspects of the loaves [16,17]. A previous work [18] showed a positive impact of dairy products addition, such as fresh yogurt, on gluten-containing bread where a significant improvement on functional properties and nutritional value was noticeable.

Yogurt (Yg), is considered the most popular dairy product worldwide for its nutritional and health benefits, since it is a source of protein (casein), exopolysaccharides (EPS), vitamins (B2, B6, and B12), and minerals (such as Ca, P, and K), representing an interesting alternative for new bakery products [18,19]. In this context, incorporation of fresh Yg in gluten-free bread formulations can be an interesting approach to mimic the gluten network, while improving the nutritional value of gluten-free breads.

The aim of the present work was to explore the potential of fresh yogurt addition on GF bread making and to overcome the technological challenge involved in gluten removal, improving the GFB quality. Different levels of Yg addition to a GF dough formulation, previously optimized, was tested, and the impact on dough, based on steady shear behavior and oscillatory measurements, was assessed. The effect on GFB quality, by the evolution of the crumb firmness and staling rate, during the storage time, as well as from bread quality parameters and nutritional profile, was also evaluated.

## 2. Materials and Methods

### 2.1. Raw Materials

Gluten-free bread was prepared using rice flour (composition per 100 g: Moisture content 10.9 g, protein 7.0 g, lipid content 1.3 g, carbohydrates 80.0 g, fiber 0.5 g), buckwheat flour (composition per 100 g: Moisture 13.1 g, protein 13.3 g, lipid content 3.4 g, carbohydrates 61.5 g, fiber 10.0 g, salt 0.08 g, phosphorus 0.35 g, and magnesium 0.23 g), and potato starch (composition per 100 g: Moisture content 17.5 g, protein 0.2 g, lipid content 0.1 g, carbohydrates 80.0 g).

The fresh yoghurt (Yg) used was a commercial product from LongaVida, Portugal (composition per 100 g: Moisture content 88.5 g, protein 3.7 g, lipid content 3.7 g, carbohydrates 5.5 g, fiber 0.8 g, 0.06 g of salt, and 0.12 g of calcium). The dry extract of Yg was determined from the standard Portuguese method: NP.703-1982 (Standard Portuguese Norm), corresponding to 11.5% of dry matter.

Commercial white crystalline saccharose (Sidul, Santa Iria de Azóia, Portugal), sea salt (Vatel, Alverca, Portugal), baker’s dry yeast (Fermipan, Lallemand Iberia, SA, Setúbal, Portugal), vegetable oil (Vegê, Sovena Group, Algês, Portugal), and xanthan gum (Naturefoods, Lisboa, Portugal) were also used.

#### 2.1.1. Bread Dough’s Preparation and Bread making

Gluten-free bread dough was prepared in a thermo-processor (Bimby-Vorwerk, Cloyes-sur-le-Loir, France), according to the procedure earlier described [20], with some modifications. First, the yeast was activated in warm water on the processor cup, during 2 min at position 3. The rest of the ingredients were added and homogenized during 1 min at position 6, and the kneading was carried out during 10 min. Hydroxy-propril-methyl-cellulose (HPMC) was replaced by xanthan gum and the fermentation time was reduced to 20 min. Preliminary assays showed that 20 min of fermentation was enough to obtain expanded doughs ready to bake. Further fermentation time led to dough structure breakdown, losing the gas (CO_2_) produced and retained during this bread making step. Bread baking conditions: Oven with convection at 180 °C during 30 min.

Different GFB formulations tested are summarized in Table 1. Other ingredients added were kept constant: Salt, 1.5%; sugar, 2.8%; dry yeast, 2.8%; xanthan gum, 0.5%; and vegetable oil, 5.5%.

Fixing the viscosity torque values of the gluten-free control dough (16.0 ± 2.0 milliNewton meter - mNm), previously optimized, the water flour absorption was determined for each Yg bread formulation tested (based on 14% of moisture basis), considering the water coming from Yg additions, applying the mixing curves procedure by MicrodoughLab assays (Perten Instruments, Hägersten, Sweden). Control dough and doughs obtained with Yg incorporations, were prepared considering the levels 10, 20, 30, and 40 g of Yg into dough, corresponding to 5% up to 20% *w*/*w* (weight/weight) in overall percentage. Replacements were based on gluten-free flours basis, substituting the dry extract of each Yg percentage on 100 g of flour [18].

#### 2.1.2. Dough Rheology Measurements

All rheological measurements were carried out in a controlled stress rheometer (Haake Mars III—Thermo Scientific, Karlsruhe, Germany), with a universal temperature control Peltier system to control temperature, using a serrated parallel plate sensor system (PP35 and 1-mm gap), to overcome the slip effect [21,22].

The rheology features of the dough were evaluated after 20 min of fermentation time, and all the assays were conducted at 5 °C of temperature to inactivate the yeast fermentative activity.

The impact of the yogurt addition on dough viscosity behavior was assessed by flow curves, under shear steady conditions ranging the shear rate from 1.0 × 10^−6^ to 1.0 × 10^3^ s^−1^.

The Carreau model was used to model the flow curves obtained, applying the Equation (1):η = η_0_/[1 + (γ***^.^***/γ***^.^***_c_)^2^] ^s^(1)
where η is the apparent viscosity (Pa s), γ is the shear rate (s−1), η_0_ is the zero-shear rate viscosity (Pa s), γ**^.^**_c_ is a critical shear rate for the onset of the shear-thinning behavior (s−1), i.e., the value corresponding to the transition from Newtonian to shear-thinning behavior, and s is a dimensionless parameter related to the slope of this region.

Frequency sweep was applied to evaluate the impact of the Yg addition on dough structure, and the evolution of the viscoelastic functions, storage (G’) and loss (G’’) moduli, were obtained ranging the frequency from 0.001 Hz to 100.0 Hz, at a constant shear stress (10 Pa), within the linear viscoelastic region of each sample, previously determined (at 1 Hz).

All rheology determinations were repeated at least three times to ensure the reproducibility of the results.

### 2.2. Quality Assessment of the Gluten-Free Breads

#### 2.2.1. Bread Firmness and Staling Rate

Bread texture was evaluated using a texturometer TA-XTplus (Stable MicroSystems, Surrey, UK) in penetration mode, according to the method earlier described [18,23].

Comparison of the bread texture, with different contents of Yg, was performed in terms of firmness, and the staling rate of the breads was evaluated, measuring the firmness during a storage time, during 96 h (4 days).

Staling bread rate was described as a function of the Yg incorporation by a linear Equation (2)
Firmness = A × time + B,(2)
where A can be considered the staling rate and B the initial firmness of the bread.

#### 2.2.2. Quality Parameters of the Gluten-Free Bread

Resulting gluten-free breads were evaluated based on after baking quality parameters, such as moisture, water activity (aw), bake loss (BL), and specific bread volume (SBV) (cm^3^/g), as earlier described [18].

Bread moisture was determined according to the standard method (American Association of Cereal Chemists (AACC 44–15.02)). Water activity (aw) variations were determined at room temperature (Hygrolab, Rotronic, Bassersdorf, Switzerland). Bread volume was measured by the rapeseed displacement standard method AACC 10-05.01, after two hours of bread cooling down. Specific volume (cm^3^/g) was calculated as the ratio between the volume of the bread and its weight. Weight loss during baking (baking loss) was assessed by weighing the bread forms before and after baking. These measurements were carried out in triplicates.

Bread crumb color was recorded using a Minolta colorimeter (Chromameter CR-300, Minolta—Osaka, Japan) after calibration with a white calibration plate (L* = 97.21, a* = −0.14, b* = 1.99). The data collected from three slices of each bread measured at three different locations of the slices were averaged and expressed using illuminative D65 by L* a* b* scale, where: L* indicates lightness, a* indicates hue on a green (−) to red (+) axis, and b* indicates hue on a blue (−) to yellow (+) axis.

### 2.3. Nutritional Composition of the Gluten-Free Breads

Nutritional profile characterization of the gluten-free breads was based on protein (International Organization for Standardization (ISO-20483:2006)), lipids (NP 4168), ash (AACC Method 08-01.01), carbohydrates (calculated by difference), and total minerals contents (ICP-AES-Inductively Coupled Plasma-Atomic Emission Spectrometry: Thermo System, ICAP-7000 series) as described in a previous study [18]. All the experiments were performed in triplicate.

### 2.4. Statistical Analysis

The experimental data were statistically analyzed by determining the average values and standard deviation, and the significance level was set at 95% for each parameter evaluated. Statistical analysis (RStudio, Version 1.1.423, Northern Ave, Boston) was performed by applying variance analysis, the one factor (ANOVA), and post hoc comparisons (Tukey test). Experimental rheology data was fitted to nonlinear Carreau model, using the TA Instruments’ TRIOS software.

## 3. Results and Discussion

### 3.1. Dough Rheology Measurements

#### 3.1.1. Steady Shear Flow Curves

The effect of the yogurt addition, at different levels, on steady shear behavior of the gluten-free doughs was evaluated, and the flow curves obtained are presented in Figure 1. It can be observed that all systems showed a typical shear-thinning behavior: An initial Newtonian region with constant viscosity at low shear rate, and as the shear rate values increase the dough viscosity began to decrease, following a straight-line decay. Similar results were obtained from the study of the different hydrocolloids’ (e.g., xanthan gum) and dairy proteins’ interaction on rheology properties of GFB formulations [17].

The experimental data presented in Figure 1 were fitted well by the Carreau model (R^2^ > 0.967), and the values of the main parameters that characterize the flow behavior are summarized in Table 2.

As it can be observed from the zero-shear rate viscosity values (η_0_)_,_ higher dough viscosities were obtained for the control dough (CD) and yoghurt dough at 5% (YgD_5%__,_ lower level of Yg tested). According to previous works [17,24], GF doughs obtained with xanthan gum (XG) addition showed high viscosities and flow behavior indexes due to the complex aggregates formed by strong molecular linkages. However, a balance must be reached since high viscosity may retain bubbles in the batter, but it may also restrict expansion during baking [25].

Nevertheless, increasing the amounts of Yg resulted in a significant decrease of dough viscosity (ɳ_0_), varying from 290.00 k Pa s (CD) to 9.50 k Pa s (YgD_20%_), representing a reduction of around 96.60%. One can suggest that the Yg incorporation promoted a dilution effect of starch–cereal protein–xanthan gum interaction density, decreasing the dough viscosity under shear conditions. In addition, the presence of the casein and of the exopolysaccharides (EPS) coming from Yg (produced by lactic acid bacteria) impacted the system, improving the lubrication and flexibility of the dough network. These effects can also be explained by the ability of EPS to bind water and retain moisture, contributing to the increase in the water-holding capacity [26] and, possibly, reducing the rigidity of the linkages between the different molecules of the GF dough matrix. Similar findings were obtained by other authors [27], investigating the addition of caseins and albumins on GFB formulations, where a considerable reduction on dough viscosity was obtained.

In terms of critical shear rate, no significant differences were observed. All the dough systems’ viscosity began to decrease around 2.0 × 10^−3^ s^−1^. This behavior suggests that, although the addition of Yg can promote the dilution effect on molecular links density as well as reducing the stiffer complex aggregates, it seems to have no effect on the breakdown of the dough matrix.

#### 3.1.2. Dough Viscoelastic Behavior

The changes on viscoelastic behavior, expressed in terms of elastic (G’) and viscous (G´´) moduli, of the gluten-free doughs, with different levels of Yg addition, were evaluated on fermented doughs, by oscillatory frequency sweep measurements.

The comparison of the mechanical spectra of control dough (CD) and doughs obtained with Yg additions (YgD) are represented in Figure 2.

Figure 2 shows that the elastic (G´) and viscous (G´´) moduli values obtained for Yg doughs, from 10% of Yg addition, were lower than control dough (CD), indicating a formation of weaker structures more like batter. No significant differences in viscoelastic profile for lower values of Yg tested (YgD_5%_) were observed, comparing with CD.

These results are aligned with those obtained by steady shear flow curves, discussed above, where a significant reduction of dough viscosity was registered from 10% on of Yg addition. Similar results were obtained by other researchers [10,28], using different gluten-free flours, proteins sources, and hydrocolloids in GFB formulations.

Analyzing in detail the results (Figure 2) at low frequencies, all the dough systems displayed a viscoelastic fluid behavior with values of G” higher than G´, characteristic of an entangled network [29,30,31], probably formed by the proteins’, exopolysaccharides’, and starch molecules’ interaction. However, with the frequency increase, the crossover of both moduli occurred and the dominance of the G´ over the G´´ was observed, expressing a typical pseudo-gel behavior [28,29,30], with high frequency dependence [32]. These results agree with previous findings obtained by the study of the rheology evolution of gluten-free flours and starches during bread fermentation and baking [33].

It can be observed that the frequency values of the G´ crossing over the G´´ were reduced by the Yg addition to dough, varying from 0.025 Hz for CD to 0.004 Hz for YgD_20%_ (higher level tested). Based on these results, and although the Yg addition promoted significant changes on dough structure, some reinforcement of the molecular linkages on the dough matrix should be considered. This may be explained by the presence of casein and exopolysaccharides coming from Yg that are acting on the system through three probable mechanisms:(1)Improving the orientation and disentanglement of the molecular linkages on the dough matrix, under oscillatory conditions [34], by lubrification and flexiblizing effects,(2)Reducing the stiffer network structure formed by xanthan gum, giving more flexibility to the dough network [17], and(3)Reinforcing the dough molecular bonds, by additional casein and exopolysaccharides interaction, giving more stability to the dough [13,17].

These results showed that the Yg additions promoted considerable changings in viscoelastic properties that resulted in the improvement of the dough network capacity to incorporate and retain gas bubbles produced by yeast´s fermentative activity, consequently, reducing significantly the viscoelastic properties of the dough under oscillatory conditions. This observed behavior resulted in better texture properties and higher specific volumes of the bread [13].

From previous work, it was already stated that interactions between protein and polysaccharides led to similar changes in the viscoelastic functions (G´ and G´´) profile of gluten-free bread doughs [10]. These findings are also in line with those obtained by other researchers [17] evaluating the effect of different hydrocolloids and proteins on rheology properties of GFB formulation.

### 3.2. Evaluation of the Gluten-Free Bread Properties

#### 3.2.1. Bread Texture and Staling Rate

The texture of the GFB produced with different contents of Yg addition was evaluated based on bread crumb firmness by a puncture test, and subsequent bread staling rate obtained from the evolution of the bread firmness, during the storage time of 96 h at room temperature [18,23].

From Figure 3, it can be observed that the Yg addition had a significant (*p* ≤ 0.05) positive impact to increase the bread crumb softness: Initial and final values of control bread firmness were higher than all levels of Yg tested, varying from 2.25 Newton (initial) to 6.50 Newton (final) for CB and 0.82 Newton (initial) to 1.35 Newton (final) for higher levels of Yg tested on breads (YgB_20%_), representing a decrease of 64% and 80% of crumb firmness values, respectively.

Bread staling rate was described as a function of time (R^2^ > 0.974). The linear parameters presented in Table 3 clearly reflect the impact of Yg additions on bread staling rate (A, the slope) and initial firmness (B, the interception).

The staling rate of the Yg breads were significantly lower than for the (CB), ranging from 0.044 N/h (CB) to 0.005 N/h for higher level of yoghurt tested yoghurt bread (YgB_20%_), representing a reduction of 90%. Similar results were obtained by other authors [35] evaluating the impact of the dairy powders on loaf and crumb characteristics and on shelf life of GFB. These findings are also in line with those obtained by a previous work [18] evaluating the impact of fresh dairy products’ addition on technological, nutritional, and sensory properties of wheat bread. Good results of bread texture and staling rate can be explained by the presence of exopolysaccharides, coming from the Yg addition, since it has been well demonstrated that EPS improve bread texture properties, and such effects are related to its ability to bind water and retain moisture, retarding the starch crystallization and, hence, the increase of bread firmness [26,36,37,38].

It can be stated that the Yg addition increased the bread crumb softness and delayed the staling rate of the GFB, leading to an increase of shelf life, which is an important industrial advantage.

#### 3.2.2. Quality Parameters of Gluten-Free Bread

The impact of the Yg addition on gluten-free quality parameters, such as crumb color, moisture, water activity (aw), bake loss (%), and specific bread volume (SBV), was evaluated. Results obtained are summarized in Table 4.

Bread crumb color was significantly changed by Yg addition, giving a lighter color (higher L* values) with more pronounced yellow tone (higher b* values) while the red tone became less dominant (lower a* values).

In terms of bread moisture, an increase in moisture content was observed (from 10% on, of Yg addition), varying from 42.0 for CB to 50.0 for YgB_20%_ (higher level tested), corresponding to an increase of 20%. Related to water activity, no significant (*p* ≥ 0.05) differences were registered for all levels tested, compared to CB.

The bake loss (BL) represented the amount of water and organic material (CO_2_ and other volatiles) lost during baking [12]. One can see (Table 4) that increasing the amounts of Yg led to a significant decrease of BL (*p* ≤ 0.05), corresponding to an increase of 27% on bread yield, comparing CB with YgB_20%_. Presumably, as discussed above, the dough network formed by the Yg addition, probably via starch molecules, vegetable protein, casein, and EPS (coming from Yg), had a higher ability to trap water and it may cause an increase in water-holding capacity, consequently, a decrease on the BL percentage [39,40,41]. These results agree with those published by other authors [26] on the combination of dairy proteins with transglutaminases on GFB formulation.

Based on specific bread volume (SBV), a significant improvement was observed by Yg addition from 10% on, varying between 1.80 cm^3^/g for CB to 2.50 cm^3^/g for YgB_20%_, representing an increase of around 40%. These improvements on the SBV were probably due to the increase in water-holding capacity, contributing to the starch molecules being more prone to form a uniform continuous starch–protein matrix, which was further enhanced during baking [12,41]. In addition, the contribution of the caseins and exopolysaccharides, coming from Yg addition, cannot be excluded, probably by building a structured starch–protein–EPS matrix [38,39], improving the flexibility of the dough network and the capacity to retain the gases produced during the fermentation process, contributing to keeping the bread structure with uniform gas cell distribution, resulting in better bread volumes than CB, as can be observed in Figure 4. 

Our results agree with those published by other authors [26] reporting the effect of dairy proteins (caseins and albumins) with transglutaminase additions on the improvement of crumb texture and better SBV.

Resuming, it can be stated that the Yg addition improved the bread quality of the gluten-free bread in all baking quality parameters evaluated.

### 3.3. Relationship between Bread Quality Parameters and Dough Rheology Properties

Bread crumb texture and volume are considered the most important baking properties by the consumers [12] and the incorporation of fresh Yg showed to be a potential ingredient to improve these bread quality parameters.

The results presented along this work suggested a relationship between bread quality parameters and dough rheology properties. Linear correlations (R^2^ > 0.9041) between the bread firmness (BF) and specific bread volume (SBV) with steady shear (dough viscosity (DV)) and oscillatory (G, elastic modulus at 1Hz) values of dough rheology were obtained. The linear correlations are illustrated in Figure 5.

As can be observed from Figure 5, while the control dough (only with xanthan gum) exhibited higher dough viscosity (DV) and elasticity (G´ at 1Hz) values, the specific volume of this bread was not high, and the crumb firmness was harder than the Yg doughs obtained. Previous works [13,17] reported that, although xanthan gum had the most pronounced effect on viscoelastic properties of the dough, the bread texture and volume were negatively affected.

In opposite, lower values of dough viscosity (DV) and elastic modulus (G´ at 1 Hz) by the Yg addition to dough resulted in breads with lower values of crumb firmness (B1) and staling rate and with higher specific volumes (B2) than control bread.

These linear correlations support the results presented along this work, showing a strong correlation between bread quality parameters and dough rheology properties. It can be stated that the dough system was improved by Yg addition, which resulted in softer breads with better volumes, as aforementioned.

These findings disagree with those obtained by other researchers [11,17] evaluating different gums and emulsifiers on gluten-free bread formulations, where higher dough viscosity and elastic values resulted in lower firmness values of the bread. It suggests that the quality of the GFB depends strongly on the type of the ingredients used and the interactions between the macromolecules into play, to mimic the structure building-like gluten matrix.

These relations between the dough rheology and bread quality can be useful to predict the behavior of the dough and provide important information for the GF bread making industry.

### 3.4. Nutritional Composition of the Gluten-Free Breads

The nutritional composition of the gluten-free breads, including the mineral profile, were determined for control bread (CB) and breads obtained with 10% and 20% of Yg addition (YgB). A significant (*p* ≤ 0.05) reinforcement on protein and ash content was observed for both levels of Yg tested. However, a remarkable effect was obtained for 20% *(w/w*) of Yg addition in both cases, representing an increase of 52% and 55%, respectively. Nutritional composition and minerals profile of the gluten-free breads are summarized in Table 5.

In terms of lipids, no significant (*p* ≥ 0.05) differences were obtained. Based on carbohydrates content, a significant decrease for higher levels of Yg tested (20% *w*/*w*) was observed, representing a decrease of 25%, due to the dilution effect of the starch content.

Regarding the minerals profile, a significant improvement (*p* ≤ 0.05) of major (Ca, K, P, Mg) and trace minerals (Fe, Cu, Mn) was obtained, representing, in general, more than 15% of the recommended daily values (Regulation (European community)), No. 1924/2006; Directive No. 90/494 (CE). Considering the higher level of Yg tested (YgB_20%_)_,_ a significant increase in Ca (158.0%), K (25.1%), P (7.0%), and Mg (4.1%) can be noticed as well as in Fe (55.3%), Cu (30.4), and Mn (37.3%), compared to control bread (Table 5). Similar results were obtained by other authors [18] testing the addition of dairy products on wheat bread.

These results showed that the Yg addition can be used to enhance the nutritional value of the GFB, increasing the amount of protein and minerals profile with a good contribution to reducing the carbohydrates intake.

## 4. Conclusions

Gluten-free bread formulations, with different levels of yogurt addition, were evaluated using dough rheology measurements and baking quality parameters.

The results of the present work showed that the functionality of gluten-free breads, in terms of bread making performances, quality parameters, and nutritional profile can be successfully improved by the addition of fresh yogurt.

Although, the yogurt incorporation significantly reduced the rheology properties of doughs, such effects resulted in significant improvements in the overall quality of the corresponding breads. Good linear correlations between bread firmness, specific volume with flow behavior, and viscoelastic functions were found, supporting the results obtained.

Resuming, the Yg showed to be a potential ingredient to improve the quality of gluten-free breads, resulting in softer breads with higher volume and lower staling rate, compared to control bread.

Related to the nutritional composition, the addition of Yg revealed to be an attractive ingredient to enhance the nutritional value of GF breads, increasing the protein and mineral contents and reducing the carbohydrates intake, with a good contribution to improve the daily diet of the celiac people.

## Figures and Tables

**Figure 1 foods-09-00111-f001:**
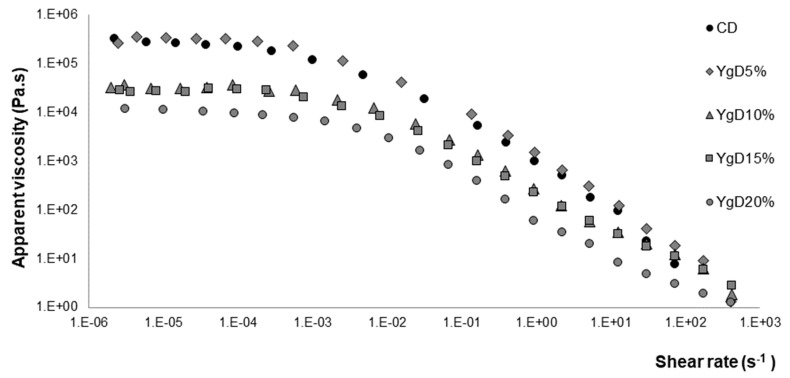
Flow curves, under steady shear conditions, obtained for control dough (CD) and doughs obtained with different levels of Yg addition (YgD).

**Figure 2 foods-09-00111-f002:**
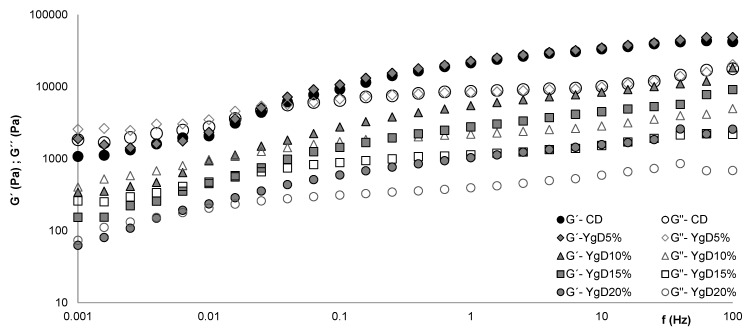
Changes in viscoelastic functions, elastic (G´) and viscous (G´´) moduli, promoted by different levels of Yg addition, YgD_5%_ up to YgD_20%_, compared to control dough (CD).

**Figure 3 foods-09-00111-f003:**
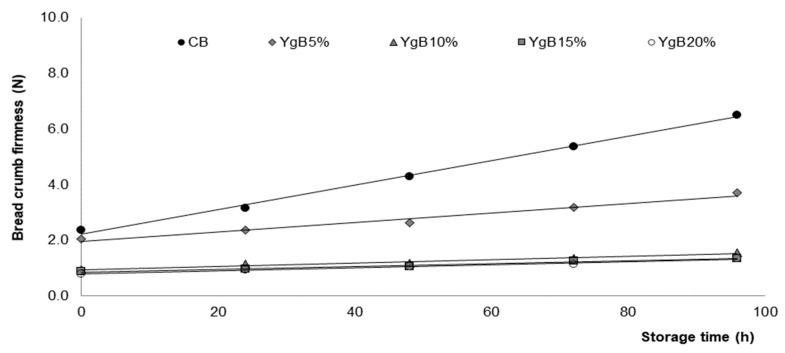
Variation of bread crumb firmness during 96 h of storage time, at room temperature, obtained for breads with different levels of Yg addition (YgB_5%_ up to YgB_20%_) compared to control bread (CB).

**Figure 4 foods-09-00111-f004:**
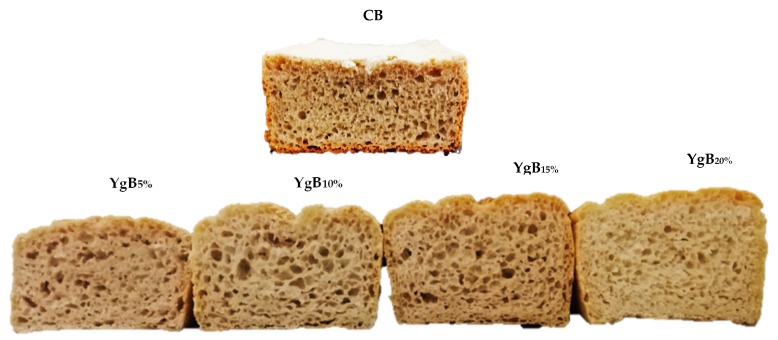
Control bread (CB) and breads obtained with different levels of Yg addition (YgB): 5, 10, 15, and 20% (*w*/*w- weight/weight*).

**Figure 5 foods-09-00111-f005:**
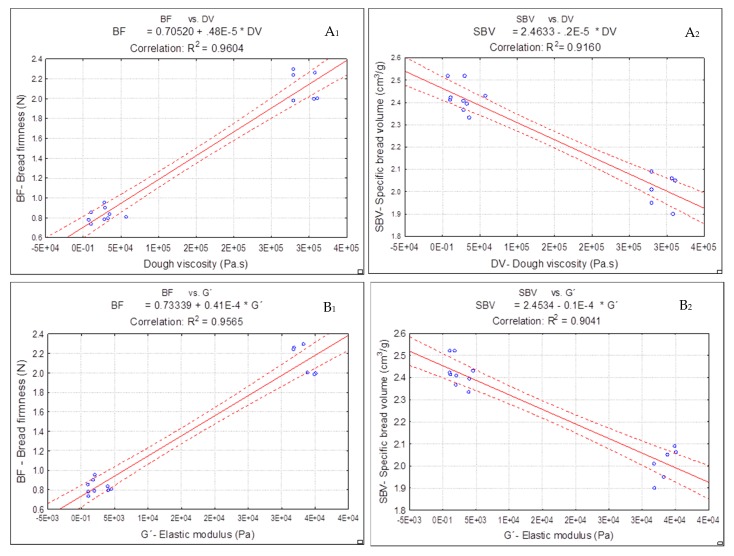
Linear correlation between bread firmness (BF) and specific bread volume (SBV) with dough viscosity (DV) and elastic modulus at 1 Hz (G´): (**A_1_**) BF vs. DV, (**A_2_**) SBV vs. DV, (**B_1_**) BF vs. G´, (**B_2_**) SBV vs. G´.

**Table 1 foods-09-00111-t001:** Gluten-free bread formulations of control bread (CB) and breads obtained with different levels of yogurt (Yg) addition (YgB), considering the dry extract coming from Yg (11.5%) to replace on flour basis.

Ingredients	CB	YgB_5%_	YgB_10%_	YgB_15%_	YgB_20%_
Buckwheat	16.6	16.4	15.3	14.4	13.3
Rice	24.8	24.6	23.0	21.5	19.9
Potato starch	13.8	13.7	12.8	12.0	11.1
Yoghurt	0.0	5.0	10.0	15.0	20.0
Added water *	37.5	32.6	31.7	30.7	31.1
Yoghurt water **	0.0	4.7	8.5	11.6	14.1
Total water absorption ***	37.5	37.3	40.3	42.3	45.2

* Determined by mixing curves performed in the MicrodoughLab equipment; ** water coming from Yg addition; *** sum of water added and water coming from Yg addition.

**Table 2 foods-09-00111-t002:** Carreau model parameters obtained for control dough (CD) and doughs obtained with different levels of Yg addition (YgD) *.

Yg levels (%)	η_0_ (k Pa s)	*γ^.^*_C_ (s^−1^)	s (slope)	R^2^
CD	290.00 ± 8.4 ^a^	1.70 × 10^−3^ ± 2.50 × 10^−4 a^	0.27 ± 0.03 ^a^	0.988
YgD_5%_	295.04 ± 13.4 ^a^	2.20 × 10^−3^ ± 1.41 × 10^−4 a^	0.23 ± 0.03 ^a^	0.980
YgD_10%_	41.90 ± 1.8 ^b^	2.52 × 10^−3^ ± 1.34 × 10^−4 a^	0.19 ± 0.03 ^ab^	0.977
YgD_15%_	28.70 ± 0.8 ^c^	2.40 × 10^−3^ ± 1.70 × 10^−4 a^	0.19 ± 0.04 ^ab^	0.976
YgD_20%_	9.50 ± 0.3 ^d^	1.43 × 10^−3^ ± 1.80 × 10^−4 a^	0.17 ± 0.06 ^b^	0.967

* Different letters (a, b, c, d) within the same column indicate significant statistical differences at *p* ≤ 0.05 (Tukey test) compared with the control bread values.

**Table 3 foods-09-00111-t003:** Bread stalling parameters: A, bread staling rate (Newton/h), and B, initial bread firmness (N), obtained for control bread (CB) and breads with different levels of Yg tested (YgB) *.

Yg Levels	B—Initial Firmness (N)	A—Staling Rate (N/h)	R^2^
CB	2.25 ^a^	0.044 ^a^	0.997
YgB_5%_	1.90 ^b^	0.020 ^b^	0.974
YgB_10%_	0.95 ^c^	0.006 ^c^	0.975
YgB_15%_	0.85 ^c^	0.005 ^c^	0.979
YgB_20%_	0.82 ^c^	0.005 ^c^	0.993

* Different letters (a, b, c) within the same column indicate significant statistical differences at *p* ≤ 0.05, (Tukey test), compared with the control bread parameters.

**Table 4 foods-09-00111-t004:** Gluten-free bread quality parameters: Crumb color (L*, a*, b*), moisture, water activity (aw), bake loss (BL), and specific bread volume (SBV) of the control bread (CB) and breads produced with different levels of Yg (YgB) addition *.

Samples	L*	a*	b*	Moisture (%)	aw	BL (%)	SBV (cm^3^/g)
CB	59.52 ± 2.85 ^a^	7.81 ± 0.23 ^a^	11.40 ± 0.20 ^a^	42.00 ± 0.33 ^a^	0.959 ± 0.006 ^a^	10.00 ± 0.41 ^a^	1.80 ± 0.08 ^a^
YgB_5%_	59.93 ± 2.83 ^a^	6.21 ± 0.08 ^ab^	13.61 ± 1.20 ^a^	43.20 ± 0.11 ^a^	0.979 ± 0.001 ^a^	9.35 ± 0.93 ^a^	2.10 ± 0.03 ^b^
YgB_10%_	62.52 ± 2.75 ^ab^	5.26 ± 0.17 ^b^	19.60 ± 0.75 ^b^	46.40 ± 0.20 ^b^	0.978 ± 0.002 ^a^	8.44 ±1.24 ^ab^	2.40 ± 0.05 ^c^
YgB_15%_	68.02 ± 1.60 ^b^	4.16 ± 0.11 ^c^	22.90 ± 0.50 ^bc^	48.00 ± 0.04 ^c^	0.981 ± 0.005 ^a^	7.50 ± 0.30 ^b^	2.43 ± 0.08 ^c^
YgB_20%_	68.60 ± 2.65 ^b^	4.05 ± 0.15 ^c^	25.32 ± 0.73 ^c^	50.00 ± 0.83 ^d^	0.985 ± 0.003 ^a^	7.30 ± 0.94 ^b^	2.50 ± 0.06 ^d^

* Different letters (a, b, c, d) within the same column, for each level of Yg, indicate statistically significant differences at *p* ≤ 0.05 (Tukey test), compared with the control bread parameters.

**Table 5 foods-09-00111-t005:** Proximate nutritional composition and mineral profile for control bread (CB) and breads enriched with 10% (YgB_10%_) and 20% (YgB_20%_) of yogurt *.

g/100 g	CB	YgB10%	YgB20%	
**Proteins**	5.34 ± 0.21 ^a^	6.90 ± 0.02 ^b^	8.10 ± 0.20 ^c^	
**Lipids**	4.83 ± 0.29 ^a^	5.20 ± 0.84 ^a^	5.60 ± 0.22 ^a^	
**Ash**	1.40 ± 0.03 ^a^	1.74 ± 0.17 ^ab^	2.12 ± 0.27 ^b^	
**Carbohydrates**	45.61 ± 1.12 ^a^	39.00 ± 2.25 ^b^	34.20 ± 1.58 ^c^	
**Kcal**	250.30 ± 1.40 ^a^	237.07 ± 2.13 ^b^	226.52 ± 2.89 ^c^	
**Minerals content mg/100 g**				**15% RDV (mg/100g) ****
**Na (g/100 g)**	0.41 ± 0.03 ^a^	0.42 ± 0.01 ^a^	0.53 ± 0.02 ^b^	
**K**	412.0 ± 8.02 ^a^	497.30 ± 5.92 ^b^	515.40 ± 16.40 ^b^	300.0
**P**	131.81 ± 5.40 ^a^	133.90 ± 0.90 ^a^	141.24 ± 2.00 ^b^	120.0
**Mg**	48.70 ± 1.02 ^a^	50.24 ± 0.56 ^a^	52.32 ± 0.82 ^b^	45.0
**Ca**	35.93 ± 2.06 ^a^	52.20 ± 0.81 ^b^	92.60 ± 2.60 ^c^	120.0
**Fe**	6.94 ± 0.12 ^a^	4.74 ± 0.67 ^b^	3.10 ± 0.05 ^c^	2.1
**Cu**	0.23 ± 0.01 ^a^	0.25 ± 0.02 ^a^	0.30 ± 0.03 ^a^	0.2
**Mn**	0.51 ± 0.01 ^a^	0.60 ± 0.02 ^a^	0.70 ± 0.03 ^b^	0.3
**Zn**	1.12 ± 0.01 ^a^	1.05 ± 0.08 ^a^	0.98 ± 0.01 ^b^	2.3

* Different letters used (a, b, c, d) indicate statistically significant differences, within the same row, at *p* ≤ 0.05 (Tukey test), compared with the bread control; ** according to the recommended daily values (RDV) established by Regulation (European Community), N° 1924/2006; Directive N° 90/494 (CE).

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
