# Peer review of "Yogurt as an Alternative Ingredient to Improve the Functional and Nutritional Properties of Gluten-Free Breads"

_foods, 2020, doi:10.3390/foods9020111_

Round 1
Reviewer 1 Report
The manuscript submitted is really interesting and provides valuable data in the field of bakery products. It is well structured and follows the path of previously published data, although dealing now with gluten-free products. Results presented are coherent, observing how an increase in yoghurt content resulted generally in lower viscosities and elastic moduli, and greater softness and lower staling rate. The section relating bread quality to rheological properties is really valuable and interesting. However, some minor issues should be considered:
Page 2, Lines74-78: please do not include salt when its content is 0g. Page 2, Line 88: please briefly summarise the procedure referenced. I cannot find the baking processing conditions (temperature, time) in the text. In Table 1, why does the yoghurt water do not match the 88.5% of the corresponding yoghurt content? (e.g. for YgB10%, why 8.5 instead of 8.8?). Page 3, Line 99: the gluten-free control dough is CB? please clarify. Page 3, Lines 113 and 128: rheological features and determinations. Page 3, Lines128: reproducibility instead of reproducible. Page 5, Lines 181-184: please do not repeat the definition of parameters of Carreau model, as they already appeared in page 3, lines 120-123. Page 6, Line 199: is it the inclusion of xanthan gum a mistake? As xantham gum is not included as ingredient in the formulation, I guess that the commented dilution effect is of starch-cereal protein interaction density, not including XG. Page 6, L218-219: Yg5% seems to display similar values than the reference (CB), just like observed in the viscosity results. Please clarify.
Best regards.
Author Response
Thank you for the revision of our manuscript and all questions/suggestion presented.
All the corrections were highlighted with pink tone.
Detailed response:
Page 2:
Line 74-78, salt content was removed.
Line 88, the procedure was briefly summarised.
Table 1, the yoghurt water presented corresponds the water coming from yoghurt addition but in over all percentage (w/w): e.g.: for 20g of Yg addition that corresponds to 10% in over all percentage (Yg10%), the water coming from 20g of Yg is 17.70 g but in over all percentage it corresponds to 8.5%.
Page 3:
line 99, gluten-free control dough corresponds to CD abbrevition. CB corresponds to control bread.
Line 128, reproducible was changed by reproducibility-
Page 5:
Line 181-184, the the definition of fitted parameters was removed.
Page 6:
line 99, xhantan gum is not a mistake, and it is included as ingredients in bread formulation, described in section 2.1.1 Bread dough´s preparation and breadmaking: Other ingredients added were kept constant: salt - 1.5 %; sugar – 2.8 %; dry yeast - 2.8 %, xanthan gum - 0.5 % and vegetable oil - 5.5 %.
The dilution effect promoted by yoghurt addittion on starch-cereal- protein interaction density also include the interaction with xhantan gum since it is part of the dough matrix developed as a fundamental thickening agent on gluten-free breads.
Line 218-219, Thanks for your important observation. It was completed and clarified on the manuscript.
Reviewer 2 Report
Excellent original topic...With gluten enteropathies on the rise, new gluten free product need to be developed. Especially those that are nutritious and flavorful
The addition of protein helps with cohesiveness and nutrition. However, need to be concerned with lactose intolerance ...may need to use lactose free yogurt
Author Response
Thank you very much for your apprecation and interesting comment/suggestion.
Reviewer 3 Report
The manuscript “Yoghurt as an alternative ingredient to improve the functional and nutritional properties of gluten free bread” is about an interesting and relevant topic in the field of celiac disease.
I think the article appears well written and the data are clearly described. The tables and the figures are clear and representative. I congratulate the authors for the quality of their research.
I have only some minor concerns that I address below:
Lines 29-30: I think that GF diet is not driving an increasing number of consumers to choose this option, but other factors are driving non-celiac people in having a GF diet (commercial, fashion…) Line 80: After “calcium” a parenthesis is lacking Lines 254-255: probably a mistake in paragraph number Line 307: just “higher ability” and not “higher more ability” Table 5: please better specify the significance of letters “a”, “b”, “c” and “d”: moreover in the comment to table 5, the author stated that a significant improvement in trace minerals, including Zn, was obtained, but in the table the value of Zn decreased in breads enriched with Yg: please verify data.
Author Response
Thanks for your revisison and appreciation of our research work.
All the corrections were highlighted with yellow tone.
Detailed response:
Page 1, line 29-30: thanks for the observation. The paragraph was changed.
Page 2, Line 80: Parenthisis were included.
Page 7, Line 254-255: It was a mistake in paragraph number. It was corrected in lines 254-255 and the rest of the document until concluion section.
Page 9:
line 307: " Higher more ability" was corrected to " higher ability";
Table 5: Significance of letter was better specified on table legend;
Related to Zn content, it was the unique trace mineral that decreased with yg addition. It was corrected.